# PV Tracking Design Methodology Based on an Orientation Efficiency Chart

**José Ruelas \***, **Flavio Muñoz**, **Baldomero Lucero and Juan Palomares**

Departamento de Ingeniería Mecánica, Instituto Tecnológico Superior de Cajeme, Cd. Obregón 85024, Mexico; fmunoz@itesca.edu.mx (F.M.); blv@itesca.edu.mx (B.L.); jepalomares@itesca.edu.mx (J.P.)

**\*** Correspondence: eruelas@itesca.edu.mx; Tel.: +55-644-418-8650

**Abstract:** This work describes a new photovoltaic (PV) sun tracker design methodology that utilizes the advantages that the orientation and efficiency of the PV panel offer due to the latitude of the installation zone. Furthermore, the proposed design methodology is validated experimentally via the implementation of a solar tracker with dual axes at a specific location (27.5° latitude). In this case, the methodology enables the incorporation of a high-availability, low-accuracy, and low-cost tracking mechanism. Based on the results, the feasibility of this type of solar tracker for latitudes close to 30° is demonstrated, as this tracking system costs 27% less than the traditional commercial systems that use slew drives. This system increases the collection efficiency by 24% with respect to a fixed device. The proposed methodology, which is based on an orientation efficiency chart, can be applied to the construction or control of other types of solar tracker systems.

**Keywords:** design methodology; two-axis PV tracker; technician feasibility; high efficiency

## 1. Introduction

Different variants of photovoltaic (PV) systems, such as PV concentration [1], maximum power point tracking (MPPT) [2], and/or solar tracking systems, are frequently studied to increase the collection efficiency of PV systems. These systems are divided into two types—passive and active tracking systems. Passive tracking systems exhibit low resistance to wind action, feature low installation and maintenance costs, and increase collection efficiency by up to 23% [3]. Active tracking systems can be single- or dual-axis systems. Single-axis systems can increase the collection efficiency by 12–25% by following the solar trajectory along one axis, i.e., horizontal, vertical, polar, or tilted [4]. Active dual-axis solar trackers have been reported to increase the collection efficiency by 17.1–31.1% [5]. In general, dual-axis sun tracking systems can rotate around either the polar and solar declination axes or the azimuthal and elevation axes. To achieve this design, ring-rail-type structures, which are constructed to support very large PV systems subjected to strong winds [6], can be mounted on pedestals or central support structures that incorporate linear actuators for polar tracking and a slew drive for azimuthal tracking [7,8]. In addition, prototypes of other dual-axis solar tracker variants have been presented. For example, solar trackers that incorporate robotic actuators have been proposed [9–11], and a solar tracker that incorporates a complex four-bar mechanism for solar tracking has been presented [12]. To improve the accuracy and performance of dual-axis solar trackers, electronic monitoring systems for MPPT in tandem with solar tracking [13] have been incorporated. Other approaches have incorporated different control variants, such as tracking algorithms and techniques based on sensors, astronomical equations, diffuse logic, neural networks, time-based tracking systems, and Petri networks, according to recent state-of-the-art reviews presented in [14,15].

The incorporation of low-cost control methods simplifies the tracking systems. Possible methods include those presented in [16,17] and strategies for tracking systems by defining three points spaced

according to the sun's trajectory, either daily or annually [18–21], or hourly tracking [22]; however, these systems do not guarantee that an adequately high range of efficiency will be maintained. Therefore, the present work proposes a new methodology for the development of a solar tracking system that exploits the advantage offered by knowing the degree of latitude of the installation site to allow the incorporation of low-accuracy, low-cost control mechanisms that keep the PV modules in adequate ranges of efficiency as a function of orientation (EFO).

## 2. Materials and Methods

The goal of this investigation is to propose a methodology that maintains the PV value within an adequate range of EFO by developing a strategy for designing a low-precision, high-efficiency PV tracking system mechanism and facilitating the instrumentation and control required for this prototype. This proposed methodology is based on the effect of misalignment error with respect to PV efficiency [23] and the Norton design methodology [24]; complementary modifications of the process, directions for the developer, as well as instrumentation and control requirements that are based on the knowledge of and devices available to the developer are made using the sequential steps presented in Figure 1.

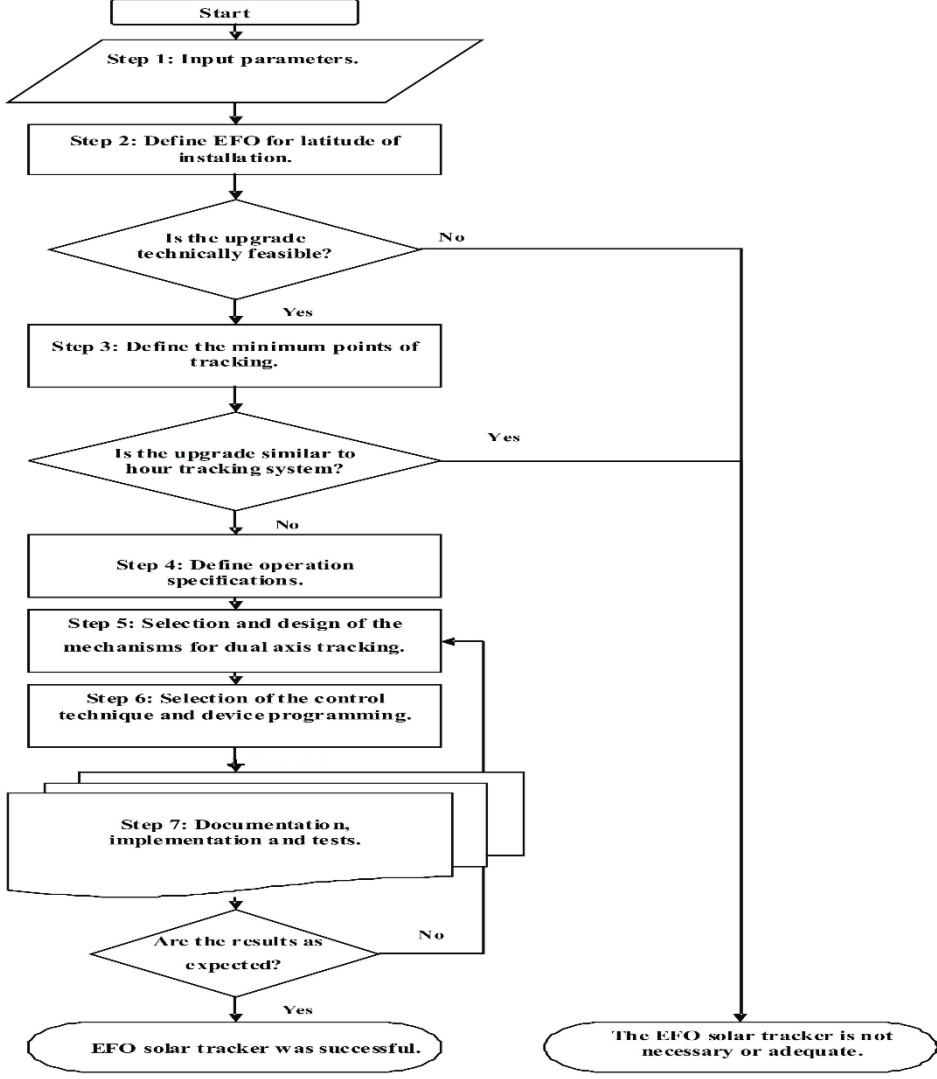

**Figure 1.** Efficiency as a function of orientation (EFO) solar tracker methodology.

The detailed description of the flow diagram shown in Figure 1 is presented below, starting with the input parameter section.

### 2.1. Input Parameters

This step defines the parameters and design rules, including the latitude of installation established according to a global positioning system (GPS) or maps, available technology, and equipment resources, for a new solar tracker. This information was obtained from manufacturers, suppliers, commercial equipment sellers, and developers. The EFO solar tracker prototype is installed at a site located at 27.5° latitude and 109° longitude. In addition, the prototype must meet the following design criteria: low cost, low maintenance, high collection efficiency, and improved performance against wind action. The specific parameter values are presented in Table 1.

**Table 1.** Solar tracker design parameters.

| Design Parameters | Value |
| --- | --- |
| Latitude | 27.5° |
| Efficiency as a function of the orientation (EFO) | 95–100% |
| Maximum wind speed | 33.3 m/s |
| Capacity | 1 kW |
| Cost | Lowest available |

### 2.2. EFO Chart of Installation Latitude

For a specific latitude, it is necessary to evaluate whether designing a new solar tracker with low precision but high efficiency is possible or whether this strategy must be replaced with another design method, as described in [14]. For this purpose, determining the PV efficiency loss due to misalignment error is necessary, using an EFO chart for the specific latitude ($\varphi$). Equations (1) and (2) are used to determine the EFO in relation to the inclination ($\beta$) and azimuth ($\alpha$) alignments [25] to ultimately determine the efficiency loss due to misalignment.

$$EFO = 100 - 100 \times \left[1.2 \times 10^{-4}(\beta - \varphi + 10)^2 + 3.5 \times 10^{-5}\alpha^2\right] \quad for \quad 15^o < \beta < 90^o \quad (1)$$

$$EFO = 100 - 100 \times \left[1.2 \times 10^{-4}(\beta - \varphi + 10)^2\right] \quad for \quad \beta \leq 15^o \quad and \quad \alpha = 0^o \quad (2)$$

An EFO graphic representation for 27.5° latitude is shown in Figure 2. For this case, the proposed EFO facilitates the incorporation of mechanisms with low resolution that can maintain the tracking errors along the azimuthal axis within the range of ±30° and the tracking errors for the tilt axis within the range of 25° while maintaining the collection efficiencies within the range of 95–100% EFO.

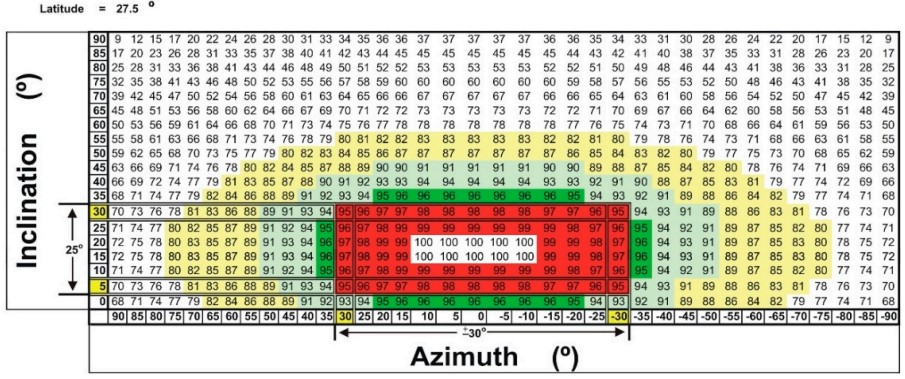

**Figure 2.** EFO chart for a photovoltaic (PV) system installed at 27.5° latitude.

### 2.3. Points and Trajectory for Solar Tracking

This section establishes the minimum number of tracking points that can be managed each day according to the EFO range established in the previous step, as well as the verified azimuth and elevation angle ranges based on the solar trajectory at the installation site throughout the year, according to the flow diagram shown in Figure 1. Assuming that the sun advances at 15° per hour, according to the EFO chart, the azimuth tracking can have a range of ±30° of the defacement. In hourly solar tracking from 6:00 to 18:00, the tracking can be divided into three fixed positions with evenly spaced angular points and time locations, as listed in Table 2. Regarding the adjustment of the solar tiles, a follow-up via daily adjustments based on the declination is proposed; this method considers that the declination value is 16°, that it can be maintained within 25° (per the solar chart for a latitude of 27.5° north), and that the mechanism has a low cost.

**Table 2.** Azimuth schedule of tracking.

| Position | Schedule of Tracking (Hours) | Azimuth (°) |
|---|---|---|
| P1 | 06:01 and 10:00 | 30 |
| P2 | 10:01 and 14:00 | 90 |
| P3 | 14:01 and 18:00 | 120 |

### 2.4. Operation Specifications

The results of this step are documents and/or pictures that describe in detail the events and actions that occur over time in the prototype solar tracker operation; the details may include labels of possible sensors and actuators needed for the instrumentation and control of the new solar tracker. The solar tracking system follows the trajectory of the sun using a three-point motion for azimuthal tracking with a clockwise open-loop control system that operates from 6:00 to 18:00 each day (Figure 3), and a linear actuator is used for tracking the declination over the year.

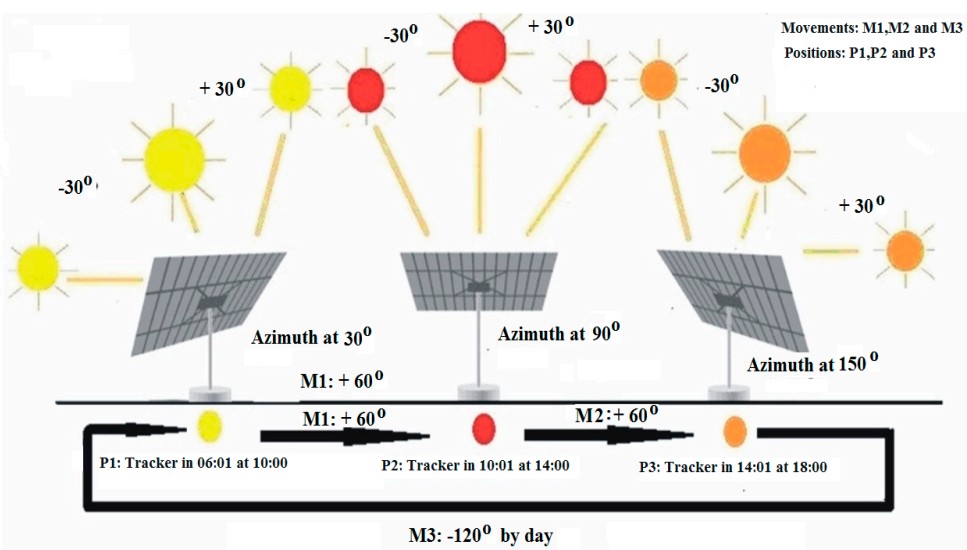

**Figure 3.** General operation of the solar tracker prototype.

### 2.5. Selection and Design of the Mechanisms for Dual-Axis Tracking

This step consists primarily of the selection of the mechanism, in accordance with the capacities required for the solar tracker prototype, using comparison matrices that facilitate the selection of devices according to the evaluation criteria. Here, "H" is assigned to high values, "M" to medium values, and "L" to low values. In addition, calculations including arithmetic, finite element analysis (FEA), and computer-aided design (CAD) tool computations are performed on the structures and

mechanisms required to develop the prototype. The first component of the design for the solar tracker is the PV system support structure, which corresponds to a 1 kW PV system composed of four panels with dimensions of 0.95 × 1.05 m and a power of 250 W. This supporting structure is designed using CAD software and FEA. FEA to determine which structure exhibits better resistance against wind action is performed using a wind speed of 33.3 m/s, which was chosen based on the maximum wind speed records at the installation location [26]. According to the FEA, the most appropriate structure uses structural steel (PTR-14, PTR-20) and $\frac{1}{2}''$ tubing. Moreover, the authors of [8] analyzed and established a better location of the azimuth as shown in Figure 4.

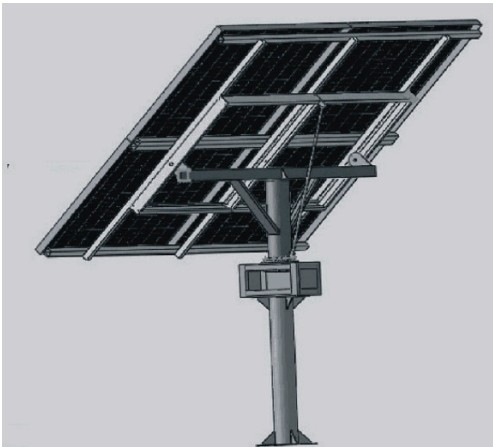

**Figure 4.** Proposed structure for the solar tracker.

Next, the tracking mechanism is selected based on the decision matrix presented in Table 3. The most viable option is a direct current (DC) gear motor adapted to a transmission because of its low cost and high availability.

**Table 3.** Actuator selection for mechanism tracking.

| Mechanism | Cost | Availability | Maintenance |
|---|---|---|---|
| Gear motor and linear actuator | L | H | M |
| Two slew drives | M | M | L |
| Two indexed motors | H | L | H |

The transmission is designed (Figure 5) to achieve the proposed resolution without affecting the previously proposed structure. A bevel gear transmission with a total gear ratio of 8:1 that is composed of a gear-motor coupling dart (1), an upper panel support (2), and bevel gears for transverse coupling (3 and 4) is proposed. The transmission uses A36 steel due to its mechanical resistance.

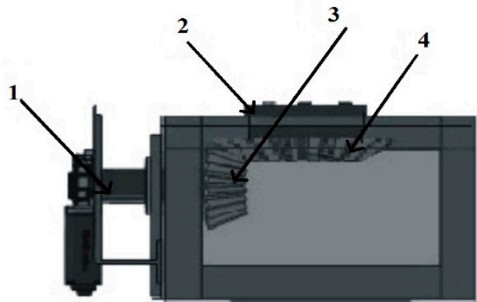

**Figure 5.** Proposed azimuthal solar tracker mechanism.

According to the calculations for a gear motor and linear actuator (such as the example shown in Figure 6, which is a worm-rack type system with a torque of 80 N-m coupled to a transmission with a step ratio of 12 and is 0.2 m in diameter), it is possible to withstand the torsion moments caused by wind action over the PV system area.

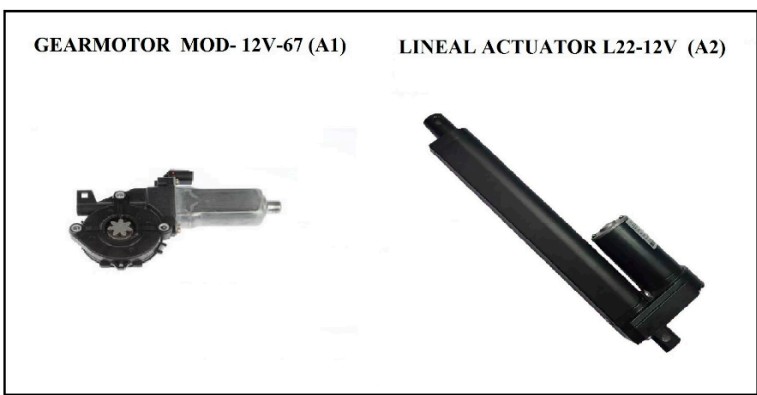

**Figure 6.** Illustration of selected actuators.

## 2.6. Selection of the Control Technique and Device Programming

The control technique (e.g., Grafcet, Petri Nets, Neural Networks, or Diffuse Logic) is selected based on the process to be performed, while considering the detailed description of the solar tracker prototype operation (Step 4) and the control device. The latter could be a field programmable gate array (FPGA), programmable logic controller (PLC), microcontroller, or industrial personal computer (PC), based on the previously selected technique. Finally, a program is developed for the algorithm and control technique previously established in Steps 4 and 5. According to the data previously obtained to fulfil the design requirements of minimizing the amount of movement and utilizing ±30° windows for maximum efficiency, a general time tracking system with an open-loop control with both automatic and manual control is proposed (Figure 7).

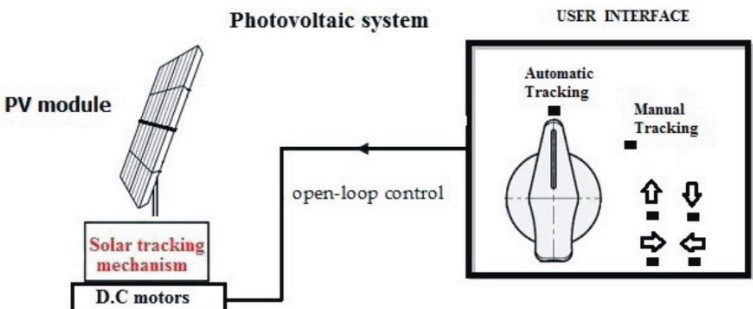

**Figure 7.** Proposed solar tracking control.

The device that will be installed on the solar tracker to perform the proposed control of A1 and A2 (DC gear motor) must be selected. A controller with the capacity to manage dates and hours and perform some arithmetic operations is required. An FPGA, a PLC, a microcontroller, and an industrial PC are considered and compared in Table 4.

**Table 4.** Control device selection matrix.

| Device | Cost | Availability | Maintenance |
|---|---|---|---|
| Diligent field programmable gate array (FPGA) for Linux | L | H | L |
| Arduino microcontroller | L | H | L |
| Festo (programmable logic controller (PLC)) compact unit control | M | M | H |
| Industrial personal computer (PC) with output interfaces | H | L | L |

One feasible option for the development of the control system is a system based on a microcontroller combined with an H bridge due to its low cost and high availability; however, FPGA boards are equally feasible. The selected control system is used to perform basic arithmetic operations and to facilitate the incorporation of a manual alignment system for the installation of the solar tracker, based on the schematic diagram shown in Figure 8.

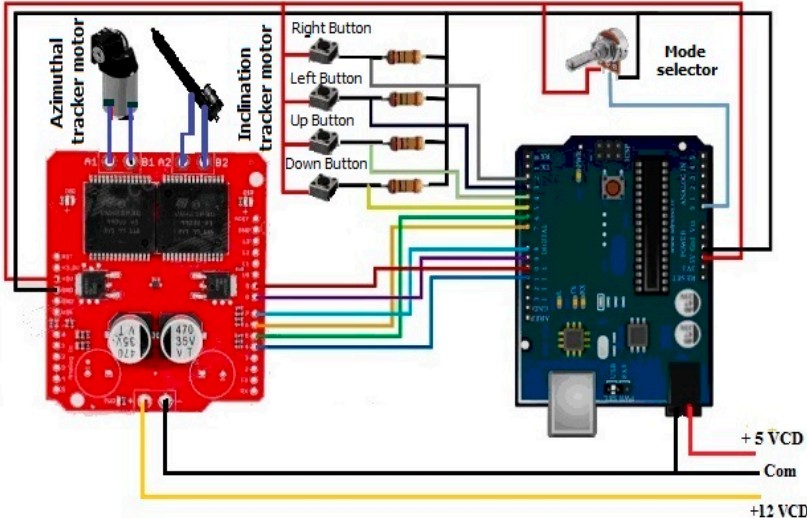

**Figure 8.** Controller connection schematic diagram.

According to the design methodology, it is necessary to program the microcontroller with an adequate technique. As detailed in Table 5, before developing the controller circuit, the control technique must be selected based on its characteristics and operation of the solar tracker prototype mentioned in the previous step.

**Table 5.** Selection of the control algorithm technique.

| Technique | Complexity | Knowledge |
|---|---|---|
| Grafcet | M | H |
| Petri nets | H | L |
| Flow diagrams | L | H |

The Grafcet technique is selected, as it facilitates programming sequences of transitions defined by a change of states and its simplicity allows the expansion of the program code for a solar tracker. In addition, the developer has a large amount of knowledge in applying this technique. Figure 9 illustrates the Grafcet technique used to develop the program of an algorithm that follows the trajectory of the sun along three points separated by 60° along the azimuth axis (Figure 5) using the low-cost microcontroller selected in the previous step that has limited digital I/O and the capacity to perform basic arithmetic operations and manage the date and time.

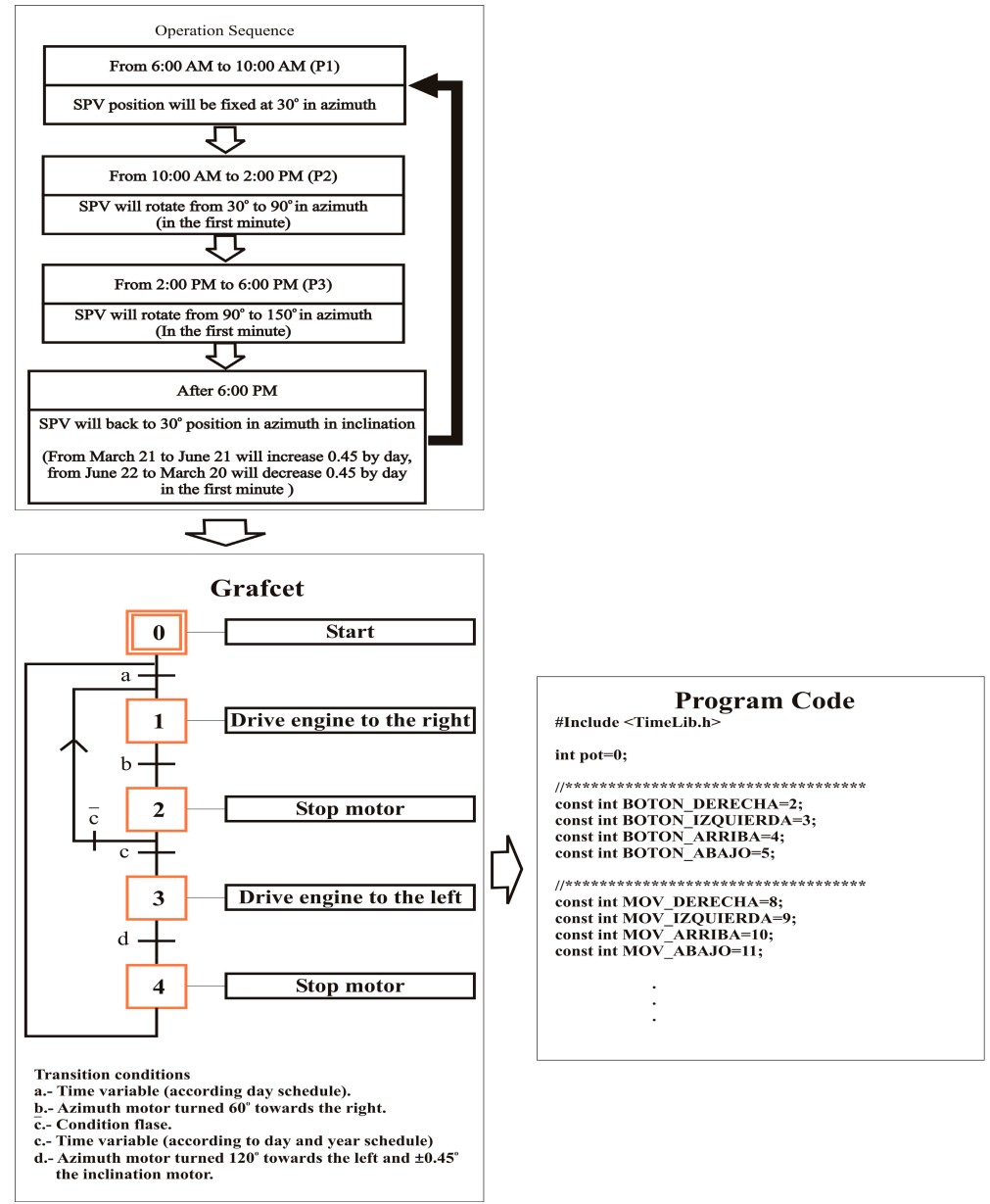

**Figure 9.** Controller programming process.

The tracking begins at point P1, located 30° above the azimuth axis, and the tracker remains at this point until 10:00. One minute later, the tracker moves to point P2, which is located 90° above the azimuth axis from the previous point, due to the activation of actuator A1, and the tracker remains at this point until 14:00. At 14:01, the tracker moves to point P3, which is located 60° from the previous point, due to the activation of actuator A1, and it remains at this point until 18:00. Finally, one minute after 18:00, the tracker returns to point P1 by activating the actuator A1 to turn 120° in the opposite direction.

## 2.7. Documentation, Implementation, and Testing

In this step, the documents required for the instrumentation, control, and construction of the prototype are collected. These are typically electrical, pneumatic, and/or hydraulic diagrams, component and prototype construction plans, instrument specifications, and program source code.

Once the plans and diagrams of the new solar tracker prototype are developed, construction of the prototype and tests are performed to determine whether the upgrade fulfils the expectations of the

EFO solar tracker prototype. If the results are inadequate, it is necessary to return to Step 5 to consider possible modifications until the results of the new solar tracker prototype are satisfactory.

### 2.7.1. Documentation

To proceed with the development of the solar tracker prototype, it is necessary to generate the following documentation: the specification of the actuators (Figure 6), the microcontroller electronic circuit diagram (Figure 8), the source code of the solar tracker control program (Figure 9), and the structural plans of the tracking system (Figure 10) in which parts P-1 and P-7 were constructed with structural steel PTR-14, P-2 to P-8 with structural steel PTR-20, and P-7 to P-10 with structural steel PTR-14 and $\frac{1}{2}''$ tubes.

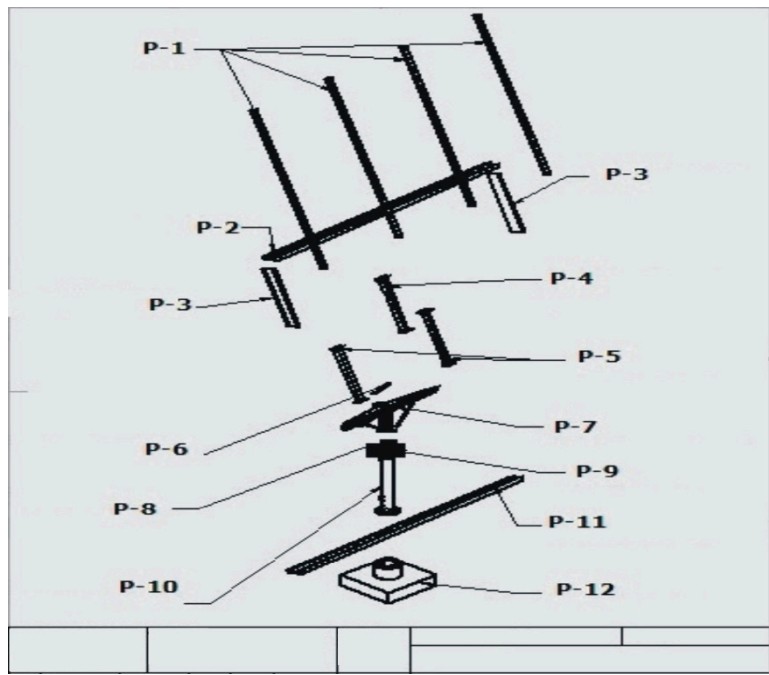

**Figure 10.** Mechanical plan for the proposed structure.

### 2.7.2. Implementation and Cost Reduction

The final implementation of the EFO solar tracker prototype is shown in Figure 11, which indicates the locations of the control system and actuators A1 and A2. The close-up view shows the details of the structure, tracker mechanisms, and control system of the solar tracker.

In this case, the application of an EFO chart to develop a solar tracker resulted in a 27% decrease in the total cost of the solar tracker compared to a dual-axis solar tracker with a mono-post and similar dimensions to the one proposed. In this investigation, the solar tracker may cost 1573 euros [27], which includes the fact that developers usually incorporate a slew drive mechanism for azimuthal tracking at a cost of 423 euros [28], while these mechanisms can be replaced by a machined system of low resolution and precision at an estimated cost of 170 euros [29].

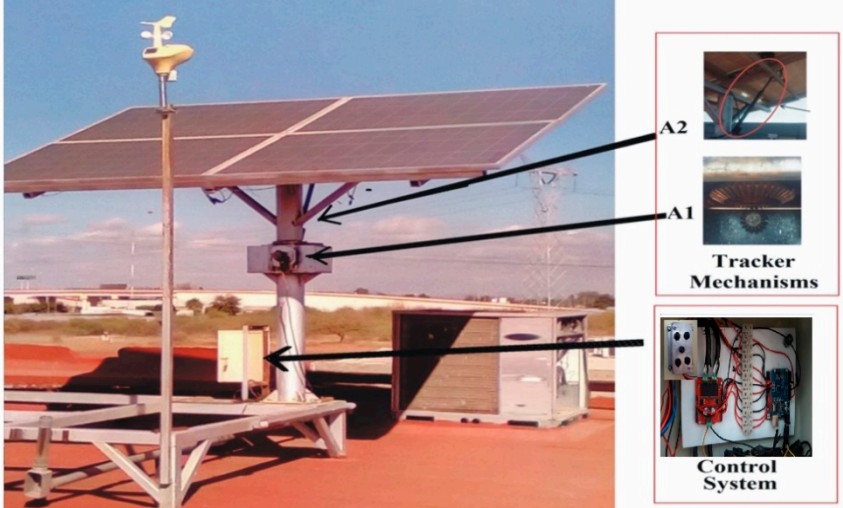

**Figure 11.** Solar tracker prototype and components.

Based on another study [30], a general dual-axis solar tracking system has an additional cost of 1.53 euros/watt with payback in 3.5–5 years. In this case, the dual-axis tracking system represents an additional cost of 1.4 euros/watt, which can be a significant cost reduction for large PV systems with a large number of solar trackers, e.g., the Nufri PV solar plant located in Lleida Spain with approximately 210 solar trackers [31].

### 2.7.3. Testing

The resistance of the solar tracker prototype to wind was tested for a specific location (27.5° latitude and 109° longitude) in 2017. During this period, the tracker remained operational and withstood maximum wind speed (v) gusts of 23.3 m/s according to data obtained from the weather station installed at the same location.

The estimated maximum torsion moment [32] caused by wind action over the PV system area (A) of 4 m$^2$ is 1306 N-m, which assumes a density of air ($\delta$) at sea level of 1.2 kg/m$^3$ at a distance (d) of 1 m and is based on Equation (3):

$$T = \frac{1}{2} \cdot \delta \cdot v^2 \cdot A \cdot d. \tag{3}$$

In the test, the solar tracker efficiency was enhanced by 23%. The results and details regarding the instrumentation and recording of the PV efficiency measurements are provided in the Results section.

## 3. Results

The energy measurement was performed by recording the power delivered by the fixed PV module and the tracking PV module, i.e., model ERDM250. The voltage and current measurements were performed using an electric circuit with a resistance of 7.5 Ohms and a load of 200 W, as shown in Figure 12. The weather conditions were recorded using an Ambient Weather WS-1001 weather station. The voltmeters are accurate to ±0.5%, the ammeter is accurate to ±2.5%, and the solar radiation recorded by the weather station is accurate to ±15%, according to the manufacturer's specifications [33,34].

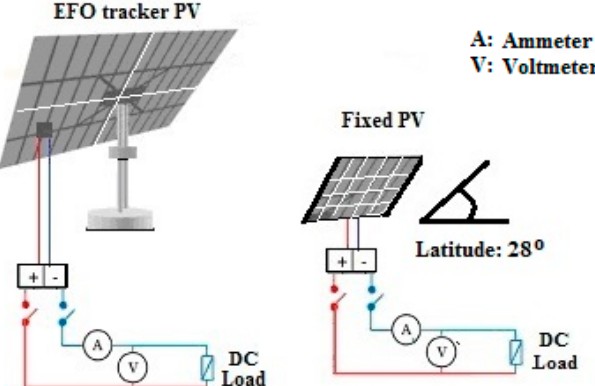

**Figure 12.** Schematic of the PV power measurement (tracking and fixed).

The measured solar irradiation ranged from 0 to 700 W/m$^2$, and the current values ranged from 0 to 5 amperes based on the PV manufacturer's current–voltage curves (Figure 13). According to the characteristics of the resistance used as the load, the voltage ranged from 0 to 30 volts. These conditions imply that the cells were operating at approximately 50% of their maximum capacity according to the manufacturer's operating curves. Additional important specifications of the PV are as follows: panel dimensions of 1640 × 990 × 50 mm, maximum power (Pmax) of 250 W, voltage at maximum power (Vmp) of 31 V, current at maximum power (Imp) of 8.08 A, short circuit current (Isc) of 8.8 A, open circuit voltage (Voc) of 37.8 V, polycrystalline type of panel, operating temperature from 40 °C to 90 °C, module conversion efficiency (hTref) of 15.39%, power temperature coefficient (bref) of −0.5%/°C, and standard testing condition of 1000 W/m^2 at 25 °C [35].

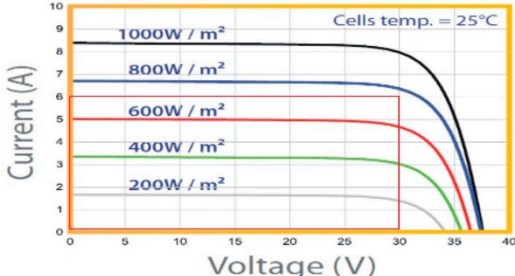

**Figure 13.** Operating range of model ERDM250 PV [35].

Measurements were taken on a specific date, which was selected based on a study by Kacira et al. [36], who found that May, June, and July feature the largest drops in efficiency due to orientation, and other studies [2,37] that evaluated the performance of a double-axis sun tracking system in comparison to a fixed PV system. The delivered energy measurements were performed using a fixed or tracking PV system of the prototype installed in Cd. Obregón, Mexico (27.5° latitude and 109° longitude) and were recorded hourly on July 12, 2018, as shown in Figure 14. These recordings can be considered adequate and congruent measurements, because during the hours around solar noon, the energy production curves of the fixed panel and the tracking panel are similar.

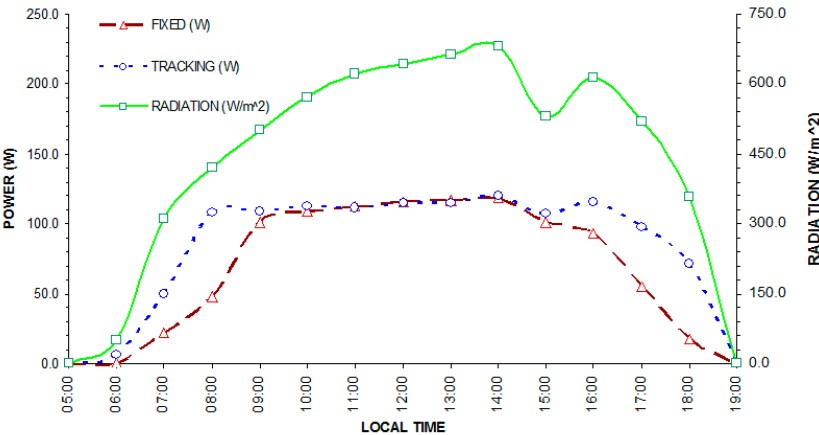

**Figure 14.** Power and production from fixed and tracking PVs.

Figure 15 shows the correlation and variance between the power measurements and incident radiation for the ERDM250 PV. The measurements have a lineal performance with a variance of ±10% of the maximum value. These values were determined based on the observations, recording measurements, and responses to electric load.

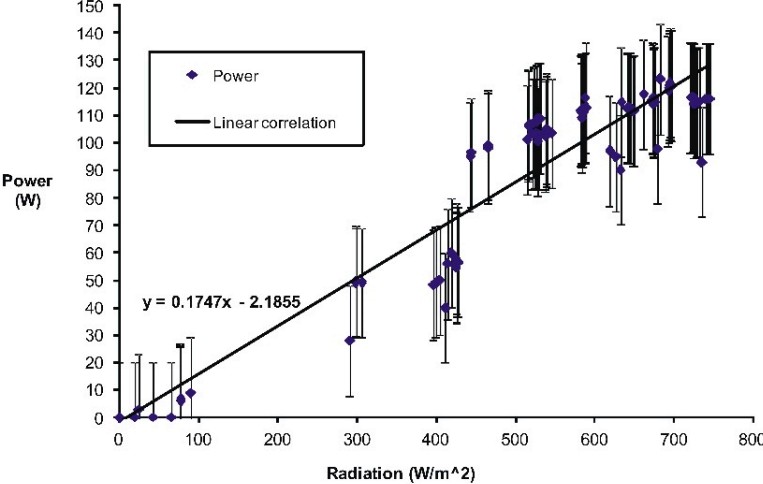

**Figure 15.** Correlation between power and incident radiation.

The EFO solar-tracking panel collected 24% more energy than the fixed PV panel. This increase in efficiency is expected to be the average value for the range of increments of dual-axis tracking systems that achieve 95–100% EFO. The increased efficiency is similar to the average increased efficiency of 24.5% for the dual-axis solar trackers that use sensors and tracking mechanisms with greater precision [5], as shown in Table 6. The increase in the technician feasibility of dual-axis PV tracking systems for latitudes close to 30° is similar to that of single-axis solar trackers for latitudes lower than 15° [38].

**Table 6.** Efficiencies of related trackers.

| Register | Type | Latitude | Test Date | Efficiency Increment |
|---|---|---|---|---|
| Al-Mohamad [7] | Two photo sensors with balance shades | Damascus, Syria 33° N | Summer of 2000 | >20% |
| Serhan and EL-Chaar [39] | Four photo sensors | Libya 33° N | - | 20–28% |
| F.M. Hoffmant [5] | Four photo sensors | Santa Cruz, Brazil 29.7° S | From June to November of 2016 | 17.2–31.1% |
| This work | Open-loop EFO tracker | Cd. Obregon, Mexico 27.5° N | July of 2018 | 23.4% |

## 4. Conclusions

This methodology enables the incorporation of a high-availability, low-accuracy. and low-cost solar tracking mechanism. Based on the results, the feasibility of this type of solar tracker for latitudes close to 30° is highlighted because this tracking system costs 27% less than the traditional commercial systems that use slew drives, while it increases the collection efficiency by 24%.

The proposed methodology can be used for developing PV trackers for different conditions, such as latitude or number of tracking points, and it enables research opportunities to apply an EFO chart to different solar tracker technologies.

**Author Contributions:** All the authors worked on designing the experiment and developing the prototype throughout the duration of the project. J.R. and F.M. created and designed the experiments. J.R. and B.L. analyzed the data, and J.R. and J.P. analyzed the documents.

**Funding:** This research was financially supported by National Technology of Mexico (grant number 067-PD).

**Acknowledgments:** The authors extend their gratitude to CONACYT and TECNM for their support.

**Conflicts of Interest:** The authors declare no conflict of interest.

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
