# Peer review of "PV Tracking Design Methodology Based on an Orientation Efficiency Chart"

_applsci, doi:10.3390/app9050894_

Round 1

Reviewer 1 Report

The Authors present an interesting method to reduce the cost of the system tracking for photoviltaic application. However, there are some important defects in the manuscript that are needed to be solved.

In general, the English writing is not clear (too long sentences without clear sense). Moreover, the explanations are limited. The content of tables and figures is not sufficient explained in the text. The information given in some figures appear too small (e.g. Figure 4 and 5). The numbering of the sections is not correct (e.g. "2.7.1" is repeated). The costs analysis is poor, and the costs reduction is one of the highlighted issues in the abstract.

I think that this manuscript has to be deeply improved in order to achieve a minimum of scientific quality. I recommend a major revision of the manuscript before considering its publication.

Author Response

Reviewer 1
1. The Authors present an interesting method to reduce the cost of the system tracking for photoviltaic application. However, there are some important defects in the manuscript that are needed to be solved.

Response: The authors thank your comments and follow the instructions to improve the article.

2. In general, the English writing is not clear (too long sentences without clear sense). Moreover, the explanations are limited. The content of tables and figures is not sufficient explained in the text. The information given in some figures appear too small (e.g. Figure 4 and 5). The numbering of the sections is not correct (e.g. "2.7.1" is repeated). The costs analysis is poor, and the costs reduction is one of the highlighted issues in the abstract.

Response:

·         Regarding the language corrections the article was sent to corrections by professionals editing service, but it is not yet available. However some corrections were made by the authors.

·         Regarding the figures was added explanations and correction on the figures (4,5,6,9 and 10)

·         The enumeration in section 2.7 was corrected.

·         Regarding the cost analysis, additional comments are added (line 226-238) and references (30-31).

Reviewer 2 Report

General speaking, the solar tracker needs to consider the ability to carry solar panels. It is difficult applied to large power scales. It is only suitable for small power scales, so the practical value is very low.

Compared with the traditional fixed PV system, the proposed system not only needs to increase the cost of the solar tracker, but also has the cost of maintenance and parts replacement. Therefore, in the long run, the actual benefits are not good.

The author should list the specifications of the single piece PV panel used, the specifications of the overall PV panel, and indicate whether the load specifications are reasonable.

As can be seen from Fig. 14, in the main period of irradiance (09:00~15:00), there is no significant difference in power generation regardless of whether it is fixed or tracking type. That is to say, considering the increase in costs, the actual power generation benefits are not obvious.

Author Response

Reviewer 2

The authors thank your comments and follow the instructions to improve the article.

1.- General speaking, the solar tracker needs to consider the ability to carry solar panels. It is difficult applied to large power scales. It is only suitable for small power scales, so the practical value is very low.

·         Response:  Was added comment (line 226-237) indicating that the EFO CHART can be more useful, if applied to large-scale photovoltaic systems such as the one mentioned in reference (31) with more than 210 solar trackers of 2 axes by example.

2.- Compared with the traditional fixed PV system, the proposed system not only needs to increase the cost of the solar tracker, but also has the cost of maintenance and parts replacement. Therefore, in the long run, the actual benefits are not good.

·         Response: Was added comment indicating that according to a study presented in [31] the 2-axis tracking systems have a cost 1.53 euros/W installed with a recovery time of 3.5 to 5 years, in our case this cost is reduced to 1.4 euros/W (line 233-237))

3.- The author should list the specifications of the single piece PV panel used; the specifications of the overall PV panel, and indicate whether the load specifications are reasonable.

·         Response: According to the comment, were added important specifications (line 264-269) of the PV panel used in the study, reference (35).

4.- As can be seen from Fig. 14,  in the main period of irradiance (09:00~15:00), there is no significant difference in power generation regardless of whether it is fixed or tracking type. That is to say, considering the increase in costs, the actual power generation benefits are not obvious.

Response: Comment was added (279-283) indicating that 10:00 to 14:00,      the same efficiency is obtained because both PV systems have the same EFO.      However, over all day the solar tracker increase the efficiency of 24%.

Round 2

Reviewer 1 Report

There are too few changes in the revised version.

The English writing still has to be improved. For instance, from lines 187 to 191.

Most figures and tables still need to be deeper described and explained in the text.

Author Response

Response to the Editor and Reviewers

Paper number: applsci-435297

Paper title: PV Tracking Design Methodology based on an Orientation efficiency Chart

Authors: José Ruelas, Flavio Muñoz, Baldomero Lucero and Juan Palomares

The authors would like to thank the area editor and the reviewers for their time and invaluable comments. We have carefully addressed all the comments. The corresponding changes and refinements made in the revised paper are summarized in our responses below.

1. - The English writing still has to be improved, for instance, from lines 187 to 191.

According to the reviewer’s observations, the article was sent to professional editing service to grammar and style correction. The details of the language corrections are shown in the attached file.

2. - Most figures and tables still need to be deeper described and explained in the text.

·         In the figure 1, connection paragraphs were added in methodology section (Lines 63-64)

·         In the figure 4 was used reference [8] to justify the design and a detailed explanation was provided. (Lines 127-128).

·         In the figure 10 was added a paragraph with the material specification used for building the structure. (Lines 221-223).

·         In the figure 12 an explanation was added, and was referred in the article. (Lines 255-256).

For all figures and tables, it was verified that they were correctly mentioned and referenced

Reviewer 2 Report

The authors have revised their manuscript according to reviewer’s comments.

Author Response

(The authors gave the same response as above.)

Round 3

Reviewer 1 Report

The authors have improved the fomal presentation of the scientific work. However, I still can see a typo, like "Grafset" in Figure 9 -it should be "GRAFCET". I recommend that the authors check once again the whole text before publication.

After that correction, I recommend its publication.